# Virulence of *Beauveria bassiana* Strains Isolated from Cadavers of Colorado Potato Beetle, *Leptinotarsa decemlineata*

**DOI:** 10.3390/insects12121077

**Published:** 2021-11-30

**Authors:** Rostislav Zemek, Jana Konopická, Eva Jozová, Oxana Skoková Habuštová

**Affiliations:** 1Biology Centre CAS, Institute of Entomology, 370 05 České Budějovice, Czech Republic; jkonopicka@seznam.cz (J.K.); habustova@entu.cas.cz (O.S.H.); 2Department of Genetics and Agricultural Biotechnology, Faculty of Agriculture, University of South Bohemia, 370 05 České Budějovice, Czech Republic; evondras@gmail.com

**Keywords:** potato, insect pest, entomopathogenic fungi, biological control, efficacy, mycoinsecticides

## Abstract

**Simple Summary:**

The Colorado potato beetle is a serious insect pest, attacking mainly potato. This pest causes severe yield loss all over the world and it is difficult to control by chemical pesticides because it quickly develops resistance to them. In our study we investigated the potential of the fungus *Beauveria bassiana*, a natural pathogen of insects, to kill adults of the Colorado potato beetle. The novelty of this study is that strains of the fungus were isolated from naturally infected adults of the pest which were collected in potato fields in the Czech Republic. A suspension of *B. bassiana* spores was applied to test beetles and their survival was observed under constant conditions. Obtained results revealed that some strains of the fungus were able to kill almost all treated beetles in 21 days and can be therefore recommended for the development of a new biopesticide. The results of this study can thus be applied in effective biological control of the most serious pest of potato.

**Abstract:**

The Colorado potato beetle, *Leptinotarsa decemlineata* (Say), is a serious, widely distributed pest of potato and other crops. This pest is able to defoliate the host plant and cause severe yield loss. Moreover, the pest quickly becomes resistant to many chemical pesticides. Therefore, the development of novel biopesticides targeting this pest is urgently needed. The purpose of this study was to obtain new strains of the entomopathogenic fungus *Beauveria bassiana* and assess their efficacy against *L. decemlineata* adults under laboratory conditions. Twelve strains were isolated from cadavers of Colorado potato beetles collected in potato fields in the Czech Republic. Test beetles were treated by suspensions of conidia at the concentration of 1 × 10^7^ spores per milliliter and their survival was recorded daily for three weeks. The results of the bioassays revealed that all new native strains were pathogenic to *L. decemlineata* adults and caused mortality up to 100% at the end of the trial period with an LT_50_ of about 7 days. These strains were more virulent than a reference strain GHA and some of them can be recommended for the development of a new mycoinsecticide against *L. decemlineata*. Our findings also highlight the importance of searching for perspective strains of entomopathogenic fungi among naturally infected hosts.

## 1. Introduction

The Colorado potato beetle (CPB), *Leptinotarsa decemlineata* (Say) (Coleoptera: Chrysomelidae), is a serious pest of potatoes (*Solanum tuberosum* L.) in the USA and Europe as well as in Asia [1,2,3]. The biology and history of its spread have been reviewed in detail elsewhere [4,5]. Adults and larvae feed on leaves and can develop two complete generations in warm regions. Without control measures, the beetle can cause severe reductions in tuber yield or quality (tuber size) [5]. Chemical pesticides which have been used frequently for the last few decades to manage this pest are, however, often hazardous for human health as well as the ecosystem. In addition, CPB has evolved resistance to many registered pesticides [6,7,8,9,10,11] and CPB thus becomes one of the most difficult insect pests to control. Resistance of CPB to chemical pesticides, concern about their harmful effects on the environment, and increasing public demand for reduction of pesticide residues in food call for alternative, safer, yet effective control agents [12,13,14].

In addition to arthropod natural enemies of CPB, such as carabid beetles [15], lacewings larvae [16] and lady beetles [17], and besides entomopathogenic nematodes [18,19] or *Bacillus thuringiensis* [20], biological control using mycoinsecticides can be a good, environmentally friendly alternative to broad-spectrum chemical insecticides [21]. Many biopesticides based on entomopathogenic fungi (EPF) such as *Beauveria bassiana* (Bals.-Criv.) Vuill., *Isaria fumosorosea* Wize (Hypocreales: Cordycipitaceae), or *Metarhizium anisopliae* (Metsch.) Sorokin (Hypocreales: Clavicipitaceae) have been developed worldwide since the 1960s [22]. Their advantages are that they can be relatively easily produced, are able to penetrate the host cuticle so they do not need to be ingested [23], and there is no risk of resistance development in target pests and none or few side effects on non-target organisms. Moreover, synergistic combinations of microbial control agents with other technologies are expected to occur in the future [24].

The efficacy of several EPF species against CPB has been investigated in many studies. For example, in various experiments, including field trials with *Isaria farinosa* (Holmsk.) Fr. alone and in combination with other fungi, high efficacy of fungus treatment was reported in Poland [25,26], the Czech Republic [27], and in Austria [28]. Another species of entomopathogenic fungus successfully tested against CPB was *I. fumosorosea*. Strain CCM 8367 of this species isolated from *Cameraria ohridella*, Deschka and Dimić (Lepidoptera: Gracillariidae) [29,30,31] caused high mortality against CPB larvae under laboratory conditions [19]. Field trials of another study revealed that potato plots treated with *I. fumosorosea* or *B. bassiana* had significantly higher tuber yield compared to an untreated control but still lower than the yield obtained from the plots protected with chemical pesticides [32]. Similar results were obtained in small field plot experiments in the USA [33]. A study on the effect of *B. bassiana* on foliage consumption by CPB larvae revealed that the treatment reduced leaf consumption by up to 76.2%, and increasing the fungus dose reduced the larval feeding period [34]. Wraight and Ramos reported significant reductions of CPB populations after *B. bassiana* foliar treatments [35,36]. The highest sensitivity of CBP to *B. bassiana* (strain NDBJJ-BFG) was reported in the youngest larval instars using when LC_50_ was only 10^5^–10^6^ while in adults LC_50_ values were 10^7^–10^8^ [37]. A recent laboratory study of 14 Turkish isolates of *B. bassiana* showed high variability in mortality among the strains and developmental stages of CPB. Some isolates caused the highest mortality between 96.7 and 100% in the 1st and 2nd instar larvae [38] when they were sprayed with a suspension of 1 × 10^7^ conidia per mL.

The present study was aimed to assess the virulence of new strains of *B. bassiana* isolated against CPB adults. The novelty of the study is that strains were isolated from naturally infected *L. decemlineata* collected in the Czech Republic. The obtained results are intended to contribute to the development of safe and efficient biocontrol of the most serious pest of potato.

## 2. Materials and Methods

### 2.1. Beauveria Bassiana Strains

Twelve strains were isolated from naturally *B. bassiana*-infected adults of *L. decemlineata* found among hundreds of beetles collected in several commercial and experimental potato fields in South and West Bohemia, the Czech Republic, during the spring season in 2019. The beetles from different populations were kept isolated under greenhouse conditions for 1–2 weeks on potted potato plants (cv. Magda) grown from in vitro cultures in sterilized soil substrate. Dead individuals were placed on sterile wet filter paper in Petri dishes to stimulate the growth of fungi. EPF strains were isolated from CPB cadavers showing mycosis symptoms typical for *B. bassiana* infection. The strains were purified using a selective medium based on Potato Dextrose Agar (PDA) (Sigma-Aldrich, Darmstadt, Germany) with a fungicide dodine (0.05 g/L) and antibiotics cycloheximide (0.25 g/L) and chloramphenicol (0.5 g/L) (Sigma-Aldrich, Darmstadt, Germany) [39].

Species identification of the strains was verified on the basis of macroscopic, microscopic, and genetic characteristics. DNA for genetic analysis was extracted from fresh mycelium grown at 25 ± 1 °C for 7 days on Petri dishes with PDA medium. Each mycelium was collected in a sterile 1.5 mL microtube. The extraction method used was based on CTAB-PVP [40] with modification for fungi. Genomic DNA was amplified by PCR with universal primers NL1 5′-GCATATCAATAAGCGGAGGAAAAG-3′ (forward) and NL4 5′-GGTCCGTGTTTCAAGACGG-3′ (reverse) [41,42]. PCR reactions were carried out in a volume of 25 µL contained in 1× reaction buffer (75 mM Tris–HCl, pH = 8.8, 20 mM (NH_4_)_2_SO_4_, 0.01% Tween^®^ 20, 2.5 mM MgCl_2_, 200 µM dNTPs) (Sigma-Aldrich, Darmstadt, Germany), 1.25 U Taq Purple DNA polymerase (PPP Master Mix, Top-Bio Ltd., Vestec, Czech Republic), 10 pmol of both forward and reverse primer, and 50 ng template DNA. Microtubes were placed in a thermal cycler TProfessional Basic Gradient (Biometra GmbH, Göttingen, Germany) with the following program: 1 cycle of 94 °C for 5 min, 25 cycles of 94 °C for 1 min, 50 °C for 1 min, 72 °C for 1 min and 15 s, and final elongation at 72 °C for 5 min. The part of amplified PCR products was visualized on 2% agarose gel. The PCR products were sequenced by SEQme Ltd. (Dobříš, Czech Republic). The sequences obtained were edited, compiled, and aligned using Geneious (Auckland, New Zealand) software. Sequence similarity searches were performed using NCBI GenBank BLASTn.

Cultures are deposited at the Biology Centre of the Czech Academy of Sciences, Institute of Entomology, České Budějovice. GenBank accession numbers for all 12 strains are listed in Table 1. In addition, *B. bassiana* strain GHA was re-isolated from commercial mycopesticide BotaniGard^®^ WP (Certis USA, L.L.C., Butte, MT, USA) and used as a reference strain in efficacy bioassays.

### 2.2. Preparation of Fungal Suspension

*Beauveria bassiana* strains were grown on Petri dishes (ø 90 mm) containing PDA (Sigma-Aldrich, Munich, Germany, 39 g/L). The plates were incubated at 25 °C in the dark for 10–14 days. The aerial conidia were harvested by scraping them into a sterile solution of 0.05% (*v*/*v*) Tween 80^®^ (Sigma-Aldrich, Munich, Germany). The conidial suspension was filtered through sterile gauze to separate the mycelium and clusters of conidia. In the uniform suspension, the spores were counted with a Neubauer improved counting chamber (Sigma-Aldrich, Munich, Germany) and, subsequently, the suspension was adjusted to a concentration of 1 × 10^7^ conidia per mL. The suspension was left for approximately 12 h at temperature 23 ± 1 °C to accelerate and synchronize germination of conidia [43,44] before its application. The viability of spores was verified using a standard germination test [45]. Ten drops from suspension were applied using a 1 μL inoculation loop on the surface of 2% water agar, which was poured in a thin layer onto the surface of a sterile slide. After the drops had dried, the slides were moved into a wet chamber and incubated at 25 ± 1 °C for 24 h. The percentage of germinating spores was determined using an Olympus CH20 light microscope (Olympus Optical Co., Ltd., Tokyo, Japan); bright field, 400× magnification. The spore germination of all strains was >98%.

### 2.3. Bioassays

The efficacy of individual *B. bassiana* strains was tested on CPB adults collected in an organic potato field in Malonty, South Bohemia, Czech Republic. The beetles were individually immersed in the suspension of conidiospores of the fungus (1 × 10^7^ spores per mL) for 30 s (dip-test). All specimens in a control group were immersed in a sterile solution of 0.05% Tween^®^ 80 only. The beetles were then placed into polystyrene Petri dishes (vented, inner diameter 90 mm, height 15 mm, Gosselin™, Borre, France) lined with moist filter paper KA 0 (Papírna Perštejn Ltd., Pernštejn, Czech Republic) and kept under constant conditions (25 ± 1 °C and 16L:8D photoperiod). Fresh leaves from potted potato plants (cv. Magda) grown under greenhouse conditions were added daily to provide food for beetles. The filter paper was also daily moistened with distilled water to maintain optimal humidity inside the Petri dishes and replaced with a new one once a week. The insects were monitored daily for 21 days to record insect mortality and development of mycosis on cadavers. Only mycosis verified by microscopic examination of sporulating fungus as *B. bassiana* was considered. The bioassays were repeated twice; each replication tested 30 insect individuals.

### 2.4. Statistical Analysis

Cumulative mortality data were subjected to survival analysis. The Kaplan–Meier product limit estimate calculated in the LIFETEST procedure of SAS/STAT^®^ (SAS Institute, Gary, NC, USA) module [46] was used to determine both the mean and the median time to death (LT_50_, the number of days until 50% of insects were dead) for each strain. Log-rank test statistics (PROC LIFETEST) was used to test the global hypothesis that mortality (time to death) differed between strains and Šidák adjustment was used for multiple comparisons. CPB mortality and mycosis of cadavers at the end of the experiment were expressed as mean percentage ± standard error of the mean. A generalized linear model with a binomial distribution and logit link was used to analyse data. Treatment and replication were set as fixed effects. The analysis was performed using the GLM procedure (PROC GENMOD) of SAS/STAT^®^ module [46]. Means were separated by the least-square means (LSMEANS) statement of SAS with Tukey-Kramer adjustment for multiple comparisons. The analyses were performed in SAS^®^ Studio for Linux (SAS Institute, Gary, NC, USA) [47]. *p* values <0.05 were considered statistically significant.

## 3. Results

The results of bioassays showed that *B. bassiana*-treated CPB beetles survived much less compared to beetles in control (Figure 1).

Survival analysis revealed a statistically highly significant effect of strain (log-rank test, χ^2^ = 47.449, *p* < 0.0001) and pairwise comparisons found differences between GHA strain and other strains (Table 2).

As shown in Figure 1, the survival curve of GHA-treated beetles shows an almost linear decline while in other strains most beetles died during a period between 5 and 10 days after treatment, and survival curves had thus a sigmoid shape. The shortest median survival time (LT_50_ = 6.5 days) was estimated in the Bb2 strain, the longest (LT_50_ = 12.0 days) in the GHA strain (Table 2).

Cumulative mortality of *L. decemlineata* adults on the 21st day after treatment reached 26.7% in the control, while many *B. bassiana* strains were able to kill almost all treated insects (Figure 2a).

Mean cumulative mortality at the end of experiments thus ranged between 73.3 and 100%. The most virulent were particularly the strains Bb4, Bb8, Bb10 and Bb11 causing mean mortality higher than 98%. The reference strain GHA showed the lowest efficacy against *L. decemlineata* when mortality was 73.3% at the end of the trial. The effect of treatment on virulence against CPB was highly significant (χ^2^ = 169.98, df = 13, *p* < 0.0001) but pairwise comparison showed differences only between control and *B. bassiana* strains (Figure 1a). Statistically significant were also differences between replications (χ^2^ = 4.56, df = 1, *p* = 0.0327).

The average percentage of mycosed cadavers at the end of experiments in the control group was 31.3%. This indicates that about 8.3% of adults in the field population used in bioassays were naturally infected by *B. bassiana*. Depending on the strain, mycosis ranged between 66.5 and 98.3% in GHA and Bb12 strains, respectively (Figure 1b). The effect of both the treatment and replication on mycosed cadavers were statistically highly significant (χ^2^ = 70.44, df = 13, *p* < 0.0001 and χ^2^ = 54.61, df = 1, *p* < 0.0001, respectively). Pairwise comparison showed some differences among *B. bassiana* strains (Figure 1b).

## 4. Discussion

Entomopathogenic fungi as biocontrol agents against *L. decemlineata* have been investigated in many laboratory and field studies. For example, high efficacy against last instar CPB larvae was reported for *I*. *fumosorosea*, with the highest mortality reaching 93% on the 7th day after treatment of insects by suspension of blastospores with a concentration of 5 × 10^7^ spores/mL [19]. Another EPF species, *Purpureocillium lilacinum* (Thom) Luangsa-ard, Houbraken, Hywel-Jones and Samson (Hypocreales: Ophiocordycipitaceae), was found to be most effective also on the last larval instar of CPB but mortality was only 33.2% on the 10th day of treatment with a fungal concentration of 10^8^ CFU/mL [49]. Results obtained in the present study revealed that *B. bassiana*-infected adults can be frequently found in field populations of CPB. In the population used in bioassays which originated from an organic farm in Malonty about 8.3% of beetles were naturally infected. Natural infection of CPB by entomopathogenic fungi was documented by Mietkiewski et al. [50] who found *B. bassiana* to be the dominant fungal pathogen infecting about 21% of diapausing beetles in Poland. Mortality of CPB adults caused by new native strains isolated within our study reached up to 100% when beetles were treated by conidia with a concentration of 1 × 10^7^ spores/mL with an LT_50_ of about 7 days. Such efficacy is high for adult CPB beetles because it is known that different developmental stages of CPB have different susceptibility to entomopathogenic fungi, with eggs and adults being the most resistant [51,52,53]. While in the youngest larval instars the LC_50_ values for the NDBJJ-BFG strain of *B. bassiana* were 10^5^–10^6^, in adults the LC_50_ values ranged between 10^7^ and 10^8^ spores/mL [37]. The highest sensitivity of the 1st and 2nd instar CPB larvae was confirmed by a recent study when some of 14 Turkish *B. bassiana* isolates applied at a concentration of 1 × 10^7^ conidia per mL caused mortality between 96.7 and 100% while they caused lower mortality in eggs and adults [38]. Higher vulnerability of early instars towards *B. bassiana* is not unique for CPB as it has also been reported for some other insect pests [54,55].

Several field trials demonstrated a good potential of *B. bassiana* to control CPB [32,33,34,35,36,56]. Biocontrol agents are usually applied in the field as a spray, while we used dip test in our laboratory assays. Although spraying insects together with the host plant (potato leaves in this case), e.g., using a Potter tower, seems to simulate field application better, a previous study [57] revealed lower efficacy when *B. bassiana* was applied to potato leaves, and the authors, therefore, recommend developing a formulation targeting the insect cuticle rather than a formulation ensuring spores’ long attachment and survival time on the leaf surface. Moreover, several studies have documented that the persistence of EPF spores on plants is limited by many factors such as temperature, rainfall, low humidity, solar radiation, plant chemistry, host plant genotype, fungal strain, etc. [58,59,60,61,62,63,64,65,66,67,68,69].

Our results further showed little variability in efficacy against CPB adults among native *B. bassiana* strains. This could indicate that the strains are genetically similar and that the sampling site has a relatively low effect when strains are selected naturally via the same host species. Results of a recent study [38] demonstrated significant differences in virulence among *B. bassiana* isolates obtained from the soil by bait method; nevertheless, genetical analysis revealed that all tested isolates had 99% evolutionary homology with other *B. bassiana* isolates from the NCBI database.

The present study also showed that the efficacy of new native strains was significantly higher than that of the reference strain GHA which caused intermediate mortality of *L. decemlineata* adults. GHA strain, the active ingredient of mycopesticide BotaniGard^®^ WP, has been reported to be highly efficient in the control of many arthropod pests [70,71,72,73,74]. This strain was originally isolated from a chrysomelid beetle *Diabrotica undecimpunctata* Mannerheim (Coleoptera: Chrysomelidae) and host specificity thus might be one of the reasons why it was less effective compared to native strains isolated from naturally infected CPB adults. Similar results were demonstrated by Li et al. [75] who found that the *B. bassiana* strains isolated from infected mulberry longicorn, *Apriona germari* (Hope) (Coleoptera: Cerambicidae), caused the highest cumulative mortality rate and the shortest LT_50_ to the same pest species compared with other strains. The reason might be attributed to the fact that strains were isolated from their original host pest and showed a stronger ability to infect the original host compared with other strains isolated from different insects [76]. This proves that isolation of EPF from the target host might have an advantage of discovering highly virulent strains which can only germinate and expand when it infests its specific host, called host specificity [76,77,78]. The reason for host specificity is biological long-term co-evolution and mutual adaptation, which may have limited the choices of parasites for the host [77]. Genetical aspects of the host specificity were discussed, e.g., by Viaud et al. [79] and Maurer et al. [80]. In the recent study, Zhang et al. [81] sequenced and compared the genome of seventeen *B. bassiana* isolates obtained from different insect hosts including *L. decemlineata* and concluded that several mutated genes and positively selected genes may underpin the virulence of *B. bassiana* towards hosts during the infection process.

As indicated from the present study, highly virulent strains can be obtained by isolation of EPF from target pests. There are, however, other strain characteristics important for practical applications, such as spore production in mass cultures, low humidity, and UV tolerance or long shelf life of the formulated product, which need to be considered when a particular strain is selected for further steps of mycopesticide development [82,83].

## 5. Conclusions

Our findings demonstrate that most of the native *B. bassiana* strains isolated from naturally infested *L. decemlineata* adults showed high virulence against this pest and suggest that this entomopathogenic fungus could be an alternate solution to broad-spectrum chemical insecticides. Results revealed little variability among these strains collected at different sites. Their efficacy was significantly higher than reference strain GHA from BotaniGard^®^, indicating that host species from which the *B. bassiana* strain is isolated seems to be more important than the geographical origin of the strain. Further research, e.g., genetical analysis, trials under field conditions, development of effective formulations, and study on potential non-target effects, is therefore needed before some of the strains can be recommended for application as a biopesticide.

## Figures and Tables

**Figure 1 insects-12-01077-f001:**
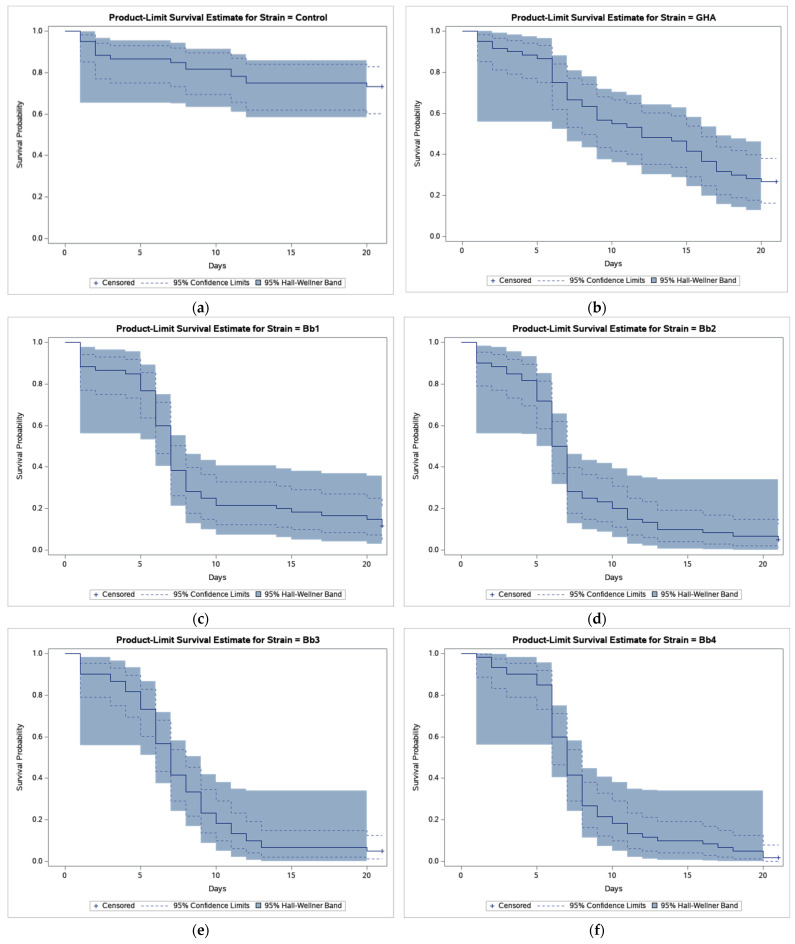
Product-limit survival estimates for *Leptinotarsa decemlineata* adults treated by *Beauveria bassiana* or distilled water with 95% confidence limits (dashed lines) and 95% Hall-Wellner bands. (**a**) Control; (**b**) Strain GHA; (**c**) Strain Bb1; (**d**) Strain Bb2; (**e**) Strain Bb3; (**f**) Strain Bb4; (**g**) Strain Bb5; (**h**) Strain Bb6; (**i**) Strain Bb7; (**j**) Strain Bb8; (**k**) Strain Bb9; (**l**) Strain Bb10; (**m**) Strain Bb11; and (**n**) Strain Bb12.

**Figure 2 insects-12-01077-f002:**
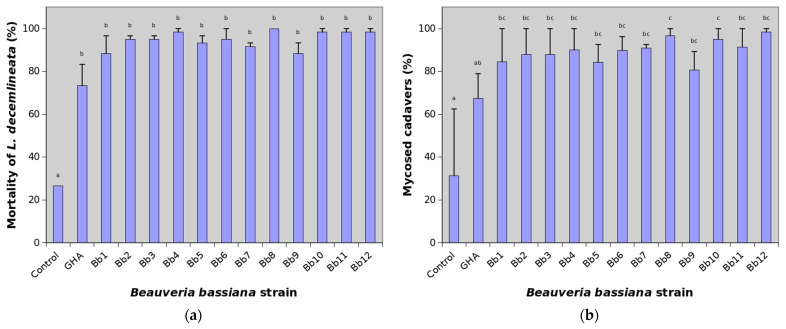
Percentage of mortality and mycosed cadavers of *Leptinotarsa decemlineata* adults treated with various strains of *Beauveria bassiana* 21 days after treatment. See Table 1 for the key to the strains. Data presented are means (± SE), with two replicates of 30 beetles for each strain. A generalized linear model was fitted and pairwise between treatment differences were tested using the least-square means. Different letters indicate significant differences between columns (*p* < 0.05). (**a**) Mortality; (**b**) Mycosed cadavers.

**Table 1 insects-12-01077-t001:** Strains of *Beauveria bassiana* isolated within this study and used in bioassays.

Strain	Sampling Site	GPS Coordinates	Genbank Accession Number
Bb1	České Budějovice	48.97417° N, 14.44867° E	MN560148.1
Bb2	České Budějovice	48.97601° N, 14.44720° E	MN749309
Bb3	České Budějovice	48.97417° N, 14.44867° E	MN749310
Bb4	Malonty ^1^	48.69105° N, 14.58950° E	MN749311
Bb5	Malonty ^1^	48.69105° N, 14.58950° E	MN749312
Bb6	Malonty ^1^	48.69105° N, 14.58950° E	MN749313
Bb7	Bělčice	49.50702° N, 13.89545° E	MN749314
Bb8	Bělčice	49.50702° N, 13.89545° E	MN749315
Bb9	Oblajovice	49.44965° N, 14.88024° E	MN749316
Bb10	Bojanovice	49.29724° N, 13.62259° E	MN749317
Bb11	Bojanovice	49.29724° N, 13.62259° E	MN749318
Bb12	Bojanovice	49.29724° N, 13.62259° E	MN749319

^1^ Organic farm.

**Table 2 insects-12-01077-t002:** Corrected mortality, mean survival time (± SE), and median lethal time (LT_50_) of *Leptinotarsa decemlineata* adults treated by suspensions of native strains of *Beauveria bassiana*.

Strain	Mortality ^1^ (%)	Survival Time ^2^ (Days)	LT_50_ (95% CI) (Days)	Log-Rank Test ^3^
Bb1	84.09	8.83 ± 0.81	7.0 (6.0–8.0)	A
Bb2	93.18	7.58 ± 0.64	6.5 (6.0–7.0)	A
Bb3	93.18	7.65 ± 0.58	7.0 (6.0–8.0)	A
Bb4	97.73	8.05 ± 0.55	7.0 (6.0–8.0)	A
Bb5	90.91	8.52 ± 0.65	7.0 (6.0–8.0)	A
Bb6	93.18	7.97 ± 0.47	7.0 (6.0–8.0)	A
Bb7	88.64	8.77 ± 0.55	7.5 (7.0–8.0)	A
Bb8	100.00	7.50 ± 0.41	7.0 (6.0–7.0)	A
Bb9	84.09	7.90 ± 0.65	7.0 (6.0–7.0)	A
Bb10	97.73	7.27 ± 0.33	7.0 (NA-NA)	A
Bb11	97.73	7.25 ± 0.43	7.0 (6.0–7.0)	A
Bb12	97.73	8.28 ± 0.43	7.0 (7.0–8.0)	A
GHA	63.64	12.33 ± 0.85	12.0 (8.0–16.0)	B

^1^ Percent of dead individuals at the end of experiment corrected for mortality in control using Abbott equation [48]. ^2^ The mean survival time and its standard error were underestimated because the largest observation was censored and the estimation was restricted to the largest event time. ^3^ Different letters indicate a statistically significant difference between lines (*p* values adjusted for multiple comparisons by Šidák).

## Data Availability

The data presented in this study are available on request from the corresponding author.

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
