# Peer review of "Virulence of Beauveria bassiana Strains Isolated from Cadavers of Colorado Potato Beetle, Leptinotarsa decemlineata"

_insects, 2021, doi:10.3390/insects12121077_

Round 1
Reviewer 1 Report
The study by Zemek et al. investigated the potential of the fungus Beauveria bassiana, a natural pathogen of insects, to kill adults of Colorado potato beetle. The purpose of this study was to obtain new strains of the entomopathogenic fungus B. bassiana and assess its efficacy against Leptinotarsa decemlineata adults under laboratory conditions.
I feel that this is a very interesting manuscript that contributes to control a very important pest that causes severe yield loss and that quickly becomes resistant to many chemical pesticides.
The manuscript is clear and I feel that it could be accepted after very minor revision. The experimental design is good; particularly I feel that the experiments were carefully performed. The analyses of data are also well performed. For me, this ms is clear and perfect as it stands, and probably I would only recommend to include a line highlighting the novelty and importance of this study.
Author Response
Point 1: The study by Zemek et al. investigated the potential of the fungus Beauveria bassiana, a natural pathogen of insects, to kill adults of Colorado potato beetle. The purpose of this study was to obtain new strains of the entomopathogenic fungus B. bassiana and assess its efficacy against Leptinotarsa decemlineata adults under laboratory conditions.
I feel that this is a very interesting manuscript that contributes to control a very important pest that causes severe yield loss and that quickly becomes resistant to many chemical pesticides.
Response 1: We appreciate very much a positive review of our manuscript.
Point 2: The manuscript is clear and I feel that it could be accepted after very minor revision. The experimental design is good; particularly I feel that the experiments were carefully performed. The analyses of data are also well performed. For me, this ms is clear and perfect as it stands, and probably I would only recommend to include a line highlighting the novelty and importance of this study.
Response 2: As suggested we included text highlighting the novelty and importance of this study in a Simple summary and at the end of the Introduction.
Reviewer 2 Report
Line 57, add not following “they do”.
Line 89, were these strains or merely separate isolates?
Line 97, was the medium potato dextrose agar?
Line 107, change containing to contained.
Line 144, need conidiophores per M.L.
Line
Line 144, how many conidiophores per ml
Line 271, change prove to proves.
Please check the references for accuracies. When latinized names are the reference title the species name is incorrectly capitalized.
.
Author Response
Point 1: Line 57, add not following “they do”.
Response 1: Thank you for revealing this mistake, we corrected it.
Point 2: Line 89, were these strains or merely separate isolates?
Response 2: We consider them as strains because they were isolated from cadavers which were expected to came from different populations of L. decemlineata.
Point 3: Line 97, was the medium potato dextrose agar?
Response 3: The medium was based on PDA, text was amended. In addition, antibiotics used were added.
Point 4: Line 107, change containing to contained.
Response 4: Text was corrected.
Point 5: Line 144, need conidiophores per M.L.
Response 5: Concentration was added as “(1 × 107 spores per mL)”
Point 6: Line 144, how many conidiophores per ml
Response 6: This information was added (see Response 5 above).
Point 7: Line 271, change prove to proves.
Response 7: Corrected.
Point 8: Please check the references for accuracies. When latinized names are the reference title the species name is incorrectly capitalized.
Response 8: We checked the references, unfortunately, latin names in titles are capitalized automatically by Zotero citation style (recent version provided by MDPI) but we can do manual changes in the final/proof version.
Reviewer 3 Report
This ms is within the scope of journal and presents interesting results. The authors tried to assess the virulence of 12 locally isolated entomopathogenic fungi against adult of Colorado potato beetle, Leptinotarsa decemlineata. The isolates were identified based on microscopic and genetic analysis – all the isolates proved effective against Colorado potato beetle.
More clarity in Methodology section is needed – it seems the results are based on single replications and this is not acceptable. Replicates within an experiment measure variability within the system, whereas repetition of the experiments ensures repeatability of the results and lack of artefacts. Would you have the same conclusions if the study was replicated is the question that should be addressed in methodology section?
As to my opinion, this article can be accepted for publication in Insects journal after minor revisions, the Editor may allow authors to revise and re-submit the ms to be considered for its publication.
Some minor points are as follows:
P1, L19: give authority for ”Leptinotarsa decemlineata”
P1, L25: replace ”specimens” with ”beetles”
P2, 53: replace ”(Balsamo) Vuillemin” with ”(Bals.-Criv.) Vuill.”
P2, L54: check taxonomy for ”Isaria fumosorosea” as ”Brown & Smith” is no more in new taxonomy
P2, L84-86: these sentences should be part of Conclusion
P3, L97: did you amend selective media with any antibiotic?
P8, L221: delete word ”on”
P8, L222: what do you mean here by ”EPN”?
P8, L230: what is word ”withing”
During review I found following relevant literature covering variety of insects, the authors may consider to cite them in Introduction or Discussion section
Tahir, T., W. Wakil, A. Ali, S.T. Sahi. 2019. Pathogenicity of Beauveria bassiana and Metarhizium anisopliae isolates against larvae of the polyphagous pest Helicoverpa armigera. Entomologia Generalis. 38: 225-242
Wakil, W., N.G. Kavallieratos, M.U. Ghazanfar, M. Usman, A. Habib, H.A.F. El-Shafie. 2021. Efficacy of different entomopathogenic fungal isolates against four key stored-grain beetle species. Journal of Stored Products Research. 93. https://doi.org/10.1016/j.jspr.2021.101845
Author Response
Point 1: This ms is within the scope of journal and presents interesting results. The authors tried to assess the virulence of 12 locally isolated entomopathogenic fungi against adult of Colorado potato beetle, Leptinotarsa decemlineata. The isolates were identified based on microscopic and genetic analysis – all the isolates proved effective against Colorado potato beetle.
More clarity in Methodology section is needed – it seems the results are based on single replications and this is not acceptable. Replicates within an experiment measure variability within the system, whereas repetition of the experiments ensures repeatability of the results and lack of artefacts. Would you have the same conclusions if the study was replicated is the question that should be addressed in methodology section?
Response 1: Thank you for this comment. In the fact, the experiments (bioassays) were conducted in two replications, each replication was done using 30 beetles. It is mentioned at the end of subsection 2.3.: “The bioassays were repeated twice; each replication tested 30 insect individuals.” This is also mentioned in Figure 2 caption.
Point 2: As to my opinion, this article can be accepted for publication in Insects journal after minor revisions, the Editor may allow authors to revise and re-submit the ms to be considered for its publication.
Response 2: We appreciate the positive evaluation and the chance to revise our manuscript.
Some minor points are as follows:
Point 3: P1, L19: give authority for ”Leptinotarsa decemlineata”
Response 3: The author name for L. decemlineata was added.
Point 4: P1, L25: replace ”specimens” with ”beetles”
Response 4: The word “specimens” was replaced as suggested.
Point 5: P2, 53: replace ”(Balsamo) Vuillemin” with ”(Bals.-Criv.) Vuill.”
Response 5: Author names were replaced.
Point 6: P2, L54: check taxonomy for ”Isaria fumosorosea” as ”Brown & Smith” is no more in new taxonomy
Response 6: It was corrected according to the valid name in Mycobank to “Isaria fumosorosea Wize (Hypocreales: Cordycipitaceae)”.
Point 7: P2, L84-86: these sentences should be part of Conclusion
Response 7: Text at the end of Introduction and Conclusions were modified accordingly.
Point 8: P3, L97: did you amend selective media with any antibiotic?
Response 8: Yes, selective media contained also antibiotics, and text was amended as “selective medium based on Potato Dextrose Agar (PDA) (Sigma-Aldrich, Darmstadt, Germany) with fungicide dodine (0.05 g/L) and antibiotics cycloheximide (0.25 g/l) and chloramphenicol (0.5 g/l) (Sigma-Aldrich, Darmstadt, Germany)”
Point 9: P8, L221: delete word ”on”
Response 9: Deleted.
Point 10: P8, L222: what do you mean here by ”EPN”?
Response 10: It was corrected to EPF. In addition, author name to P. lilacinum was added.
Point 11: P8, L230: what is word ”withing”
Response 11: It was corrected to “within”.
Point 12: During review I found following relevant literature covering variety of insects, the authors may consider to cite them in Introduction or Discussion section
Tahir, T., W. Wakil, A. Ali, S.T. Sahi. 2019. Pathogenicity of Beauveria bassiana and Metarhizium anisopliae isolates against larvae of the polyphagous pest Helicoverpa armigera. Entomologia Generalis. 38: 225-242
Wakil, W., N.G. Kavallieratos, M.U. Ghazanfar, M. Usman, A. Habib, H.A.F. El-Shafie. 2021. Efficacy of different entomopathogenic fungal isolates against four key stored-grain beetle species. Journal of Stored Products Research. 93. https://doi.org/10.1016/j.jspr.2021.101845
Response 12: Thank you for the additional literature; citations were included in the Discussion.
Round 2
Reviewer 3 Report
I have carefully gone through especially the methodology section and all suggestions throughout the ms , the authors made all the corrections and in my opinion this ms may be accepted for its publication in Insect in its present form